# Further Examination of the Psychometric Properties of the Multicomponent Mental Health Literacy Scale: Evidence from Chinese Elite Athletes

**DOI:** 10.3390/ijerph191912620

**Published:** 2022-10-02

**Authors:** Xiang Wang, Wei Liang, Jingdong Liu, Chun-Qing Zhang, Yanping Duan, Gangyan Si, Danran Bu, Daliang Zhao

**Affiliations:** 1Department of Curriculum and Instruction, The Education University of Hong Kong, Hong Kong 999077, China; 2Department of Sport, Physical Education and Health, Hong Kong Baptist University, Hong Kong 999077, China; 3Department of Physical Education, Sun Yat-Sen University, Guangzhou 510006, China; 4Department of Psychology, Sun Yat-Sen University, Guangzhou 510006, China; 5Sport Psychology Center, Hong Kong Sports Institute, Hong Kong 999077, China; 6Hubei Institute of Sport Science, Wuhan 432025, China; 7Department of Sport, Hubei University, Wuhan 430062, China; 8School of Leisure Sports and Management, Guangzhou Sport University, Guangzhou 510500, China

**Keywords:** mental health literacy, validation, validity, reliability, athletes, help-seeking

## Abstract

This study aimed to examine the psychometric properties of the Multicomponent Mental Health Literacy Scale (MMHLS) among Chinese elite athletes. Particularly, the factorial validity, convergent validity, discriminant validity, concurrent validity, internal-consistency reliability, and test-retest reliability of the MMHLS were examined. A total of 320 participants were recruited from the Guangdong provincial sports training center in China. Data collection was conducted between 30 June and 31 July 2020 using electronic questionnaires. Confirmatory factor analysis (CFA), Rasch analysis, correlation analysis and independent-sample *t*-tests were conducted using Mplus 8.3 and ConQuest 2.0. The CFA results supported the factorial validity of the three-dimensional MMHLS, consisting of knowledge-oriented MHL, beliefs-oriented MHL, and resource-oriented MHL. Cronbach’s alpha and composite reliability coefficients supported the internal-consistency reliability of the MMHLS. Moreover, the convergent and discriminant validities were supported for the subdimension of MHL-Knowledge, MHL-Beliefs, and MHL-Resources. Concurrent validity was demonstrated through correlations between MMHLS, help-seeking attitudes, and stigma. Rasch analysis provided further evidence of the psychometric quality of the instrument in terms of its dimensionality, item fit statistics, and rating scale effectiveness. Finally, test–retest reliability was 0.66 after one month. In conclusion, the 24-item three-dimensional MMHLS was verified to be a reliable and valid measurement of mental health literacy in Chinese elite athletes.

## 1. Introduction

### 1.1. Mental Disorders in Elite Athletes

Research on mental health in elite sports has grown rapidly over the last decade [1]. Considerable evidence has demonstrated a high prevalence of mental disorders (e.g., emotional distress, hopelessness, depressive symptoms) among athlete populations—particularly for those at elite levels (i.e., competing at professional, Olympic, or collegiate levels) [2,3,4]. A recent systematic review and meta-analysis has indicated that the percentage of global elite athletes who report symptoms of distress, sleep disturbance, anxiety/depression, alcohol misuse, and eating disorders/adverse eating habits is 19.6%, 26.4%, 33.6%, 18.8%, and 1–28%, respectively—which might be slightly higher than the general population [5]. In China, Tan et al. revealed that the rates of psychological problems were between 10.6% and 31.8% among elite athletes in Guangdong province [6]. Recent evidence has also indicated moderate to high levels of mental disorders in 18% of Chinese elite athletes from a sports training center [7].

Competitive sports contexts have resulted in increased pressure on elite athletes and high demands and extensive training loads pose potential threats to athletes’ mental health well-being [1,8]. Athletes usually experience not only similar mental health concerns as non-athlete peers—such as anxiety, depression and suicidal ideation, ADHD, eating disorders, and substance abuse—but also a unique range of stressors (e.g., performance expectations, frequent travel, family issues) that put them in a high-risk environment for developing or exacerbating mental health disorders [8,9]. Furthermore, incurring and recovering from physical injuries, overtraining, concussion, sleep disorders, and racial and social identity are influential factors that affect the mental health of athletes [1]. Recently, the outbreak of the COVID-19 pandemic has evidently worsened existing mental problems and increased disorder morbidities among elite athletes [2].

Despite the considerable risk and prevalence of mental disorders among athletes, coaches and athletes themselves pay more attention to sports performance—ignoring the mental health issue and its numerous negative consequences [10]. Athletes’ mental disorders can not only cause performance deterioration but can also impact their later life and future career development [1]. Based on Gardner and Moore’s stress-diathesis model [11], athletes might be sensitized to performance dysfunction due to the interaction of performance schemas, perceived skill deficits, ideographically relevant environmental events, and performance cues. Athletes with a low frustration tolerance schema typically cannot exercise sufficient self-control and frustration tolerance to achieve personal or team goals. Therefore, mental health issues directly impact elite athletes’ training, competition preparation, and performance. In addition, mental health plays a critical role in elite athlete career development because it is a significant resource for elite athletes making decisions and coping with athletic and non-athletic transitions, while mental disorders can obstruct effective decision making and transition coping [12]. Additionally, athletes with mental disorders (e.g., depression) retire earlier than healthy ones, and their symptoms might become exacerbated having a further negative influence on their social relationships, physical health, and family income [13].

### 1.2. The Role of Mental Health Literacy in the Prevention of Mental Disorders

According to previous studies on elite athlete mental health management, mental health literacy plays a crucial role in the prevention of mental disorders [7,8]. Mental health literacy (MHL) is a multifaceted concept consisting of knowledge and beliefs about mental disorders that facilitate their recognition, management, or prevention [14]. Specifically, MHL includes seven components: (1) recognition of mental disorders, (2) knowledge of how to seek mental health information, (3) knowledge of mental health risk factors, (4) knowledge of etiology/causes of mental illness, (5) knowledge of self-treatment, (6) knowledge of the professional help available, and (7) attitudes that promote the recognition of appropriate help-seeking behaviors [15]. A growing body of evidence has demonstrated that individuals who have a higher level of MHL show a lower risk/severity of mental disorders [15,16,17].

Help-seeking theory [18] provides a potential explanation for the underlying mechanisms for MHL–mental disorder associations. Help-seeking is a process and help-seeking behavior has a consistent relationship with help-seeking intentions and attitudes. It is possible to improve help seeking-behavior by comprehensively and accurately understanding mental health disorders and symptoms. More importantly, self-recognition of symptoms impacts early detection and help-seeking. Therefore, it is easier to understand why MHL benefits mental health. For example, self-recognition of symptoms is an essential aspect of MHL affecting early detection and predicting help-seeking behavior [15]. In addition, individuals who have a higher level of MHL might manage their mental health positively [15]. Previous evidence has also found that MHL is positively associated with help-seeking attitudes [17,19], and is negatively related to stigma [19,20]. For instance, a medium correlation (r = 0.42, *p* < 0.001) was found between MHL and attitudes toward mental health help-seeking among staff members working in public housing [19]. Moreover, MHL had a moderately negative relationship (r = 0.35, *p* < 0.001) with stigma for seeking professional psychological help in elite athletes [20]. In addition, previous evidence has also highlighted that MHL may differ across different demographics (e.g., gender). For example, Ratnayake and Hyde (2019) found that women reported higher MHL than men [21].

### 1.3. Measurement Instruments for MHL

A series of diverse approaches have been applied to assess MHL, and the vignette interview is the original and most widely utilized method. However, this method has several limitations, such as only focusing on one condition and only valuing the knowledge attribute of MHL, being time-consuming, and the cultural adaption of vignettes [22,23]. Such shortcomings have motivated the revision of the vignette interview method and have inspired the development of scale-based instrument tools. These types of questionnaires apply dichotomous responses (Mental Health Knowledge Questionnaire, [24]), multiple choices (Multiple-Choice Knowledge of Mental Illnesses Test, [25]), Likert responses (Mental Health Literacy Questionnaire, [26]), or a combination of the above to assess mental health literacy (Mental Health Literacy Scale, [27]). Although it is easier to use these methods for valuation, most focus on one condition instead of measuring all attributes of MHL or do not provide enough evidence of psychometric properties. Another issue with the available measurement tools is the lack of validation [23,28]. To address such issues, it is necessary to evaluate more knowledge—in particular, comparing stigma and attitudes—and to provide evidence of the psychometric properties of available scales [23].

Echoing this suggestion, Jung and colleagues developed a Multicomponent Mental Health Literacy Scale (MMHLS) based on the multifactorial construct assumption of the MHL [29]. The MMHLS includes three dimensions (i.e., MHL-Knowledge, MHL-Beliefs, and MHL-Resources) that evaluate all seven attributes proposed by Jorm et al. [14]. The factorial validity and internal-consistency reliability of the MMHLS has been well supported in employees in public housing [19]. For example, the results of a CFA indicated a good model fit: RMSEA = 0.02, CFI = 0.98, and TLI = 0.98; a reliability analysis found that the KR-20 coefficient for the whole scale was 0.83.

In addition to the general population, the MMHLS has also shown promise in measuring MHL in other contexts. For example, Sullivan et al. validated the MMHLS among collegiate athletes and their therapists, and the results supported the strong psychometric properties of the MMHLS, indicating the effectiveness of the MMHLS in assessing mental health literacy in sports populations [30]. This measurement was also validated in the male military and performed well [31]. Regarding elite athletes, as we mentioned before, there are various sport-specific factors such as overtraining, burnout, injuries, and retirement that could lead to athletes’ mental health disorders [10]. Previous studies found that almost fifteen percent of elite athletes suffered from psychological problems in some sports training centers in China [6,7]. However, compared to the variety of high-performance sports and the enormous number of elite athletes in China, there is still relatively limited research on mental health among athletes [7]—especially targeting MHL.

In China, there has been an increasing number of studies focusing on MHL during the past five years; whereas most existing studies have targeted the general population (e.g., older adults) [32], research on Chinese elite athletes is limited. With increasing concerns of mental health issues in elite athletes, a valid, reliable, and easy to operate instrument tool for MHL is necessary and would contribute to future observational and experimental studies. Given the lack of validation of the MMHLS in Chinese elite athletes, the main objective of this study is to examine the psychometric properties of the MMHLS in Chinese elite athletes—particularly including examinations of its factorial validity, convergent and discriminant validities, concurrent validity, internal-consistency reliability, and test–retest reliability.

## 2. Materials and Methods

### 2.1. Participants

This investigation was conducted with a convenience sampling of 320 elite athletes who had competed at professional, Olympic, or collegiate levels from the Guangdong provincial sports training center in mainland China. The mean age of the sample was 21.7 years (SD = 3.5). About three-fifths of the participants (57%) were men and 43% women. Most of the participants—that is, 65%—were diploma or master’s holders, while the rest had a high school education. These athletes competed in 15 sports including archery, gymnastics, diving, table tennis, weightlifting, and track and field. The majority of participants completed the course on mental health (222; 69%) and the same number of athletes confirmed that there was no mental illness patients around them.

### 2.2. Procedure

The Guangdong provincial sports training center is a cooperating agency in this project. Several sports psychologists in this article have provided long-term service for different teams in this center. To ensure a smooth investigation, the purpose and nature of the study was explained to team leaders and coaches before requesting permission to recruit participants. Upon their approval, the recruitment information was verbally disseminated to the athletes by sports psychology interns with the assistance of team coaches. Data were collected through an online survey on 30 June 2020. After 4 weeks, we evaluated the retest-reliability of the MMHLS by re-administering the questionnaire to the same sample. All the participants signed the written informed consent form before the study’s commencement. They completed the survey independently and voluntarily and were allowed to withdraw from the study at any time without any negative consequences. Each survey lasted for 10–15 min. This study was approved by the institutional review board of the Guangzhou Institute of Physical Education and was funded by the Guangdong Provincial Social Sciences Research Grant Committee (GD20XTY20).

### 2.3. Translation of the MMHLS into Chinese

Based on the protocol for scale translation, the MMHLS was translated into Chinese by two bilingual psychologists in a committee [33]. Each expert translated the scale independently and discrepancies were discussed until a consensus was reached. The translation was then validated by two additional bilingual experts using the standard back translation technique [34,35]. Subsequently, a pilot test was conducted with 15 athletes (8 male, 7 female) from 18 to 24 years old (mean = 20.9 ± 2.1 years) to check the readability and clarity of each item. Based on feedback from the athletes, mirror modifications were made to the wording and grammar.

### 2.4. Instrument

A Chinese-translated 26-item MMHLS questionnaire measured three interrelated factors of MHL, including: (1) Knowledge-oriented MHL, measured with 12 items (e.g., K6: cognitive behavioral therapy can change the way a person thinks and reacts to stress); (2) Belief-oriented MHL, measured with 10 items (e.g., B5: A person with depression will get better on his or her own without treatment); (3) Resource-oriented MHL, measured with 4 items (e.g., R3: I know where to get useful information about mental illness). The questions on the knowledge and beliefs dimensions were answered on a 5-point Likert scale (i.e., strongly disagree, disagree, neutral, agree, and strongly agree), with the option of choosing “I don’t know”, while the rest of the questions for the resources dimension were simply answered as “yes” or “no”. For the analysis, each response was grouped into two categories; for the knowledge dimension, the answers “strongly agree” or “agree” were coded as 1 and the others were coded as 0. For the beliefs dimension, the responses “strongly disagree” or “disagree” were coded as 1 and the others were coded as 0. The response format for the last five items was “yes (coded 1)” and “no (coded 0).” The Cronbach’s alpha in previous studies ranged from 0.76 to 0.84.

Attitudes toward Seeking Professional Psychological Help Scale–Short Form (ATSPPH-S, [36,37]): this scale comprises 10 items with ratings on a 4-point Likert scale ranging from 1 (totally disagree) to 4 (totally agree), where a higher score means more positive attitudes towards seeking professional help. The translated version has good reliability among elite athletes α = 0.73 [20].

The Questionnaire of Stigma for Seeking Professional Psychological Help (SSPPH, [38]) evaluates two types of stigma related to professional psychological help: public stigma and self-stigma. This scale contains ten items with a rating on a 5-point Likert scale, and a higher score indicates more stigma related to professional psychological help. In 2011, Hao and Liang used this scale to assess Chinese university students and its internal consistency reliability ranged from 0.77 to 0.81.

### 2.5. Data Analysis

Descriptive statistics and correlation analyses between study variables were performed using SPSS 27 (IBM Corp, Armonk, NY, USA). We examined the normality of the observed variables before CFA, and all results were below the cutoff values (skewness and kurtosis were 3.0 and 8.0)—which indicates that the data were closed to normally distributed. Our study had a sufficient sample size due to De Vellis suggesting a person–item ratio of between 5:1 to 10:1 for factor analysis [39]. CFA was conducted with weighted least squares means and was variance adjusted (WLSMV) due to all the indicators being dichotomous [40,41]. Model fits were evaluated using multiple fit indices, including Chi-square (𝜒^2^), the Tucker–Lewis index (TLI), the comparative fit index (CFI), the standardized root mean square residual (SRMR), and the root mean square error of approximation (RMSEA) [42]. Acceptable values of greater than 0.90 were adopted for the TLI and CFI and less than 0.08 for the SRMR and RMSEA [43,44]. We removed items with low factor loadings to improve the model fit. In addition, further details for the reliable estimation of MMHLS were provided by Cronbach’s alpha and composite reliability coefficients, which were suitable for a multi-dimensional scale to evaluate the internal-consistency reliability [45]. As the MMHLS is a three-dimensional scale, the convergent validity was examined for each dimension while the discriminant validity was examined between the three dimensions of the MMHLS (i.e., knowledge-oriented MHL, beliefs-oriented MHL, and resource-oriented MHL). The factor loadings were employed to calculate the average variance extracted (AVE) and the composite reliability (CR), where AVE > 0.50 and CR > 0.60 indicated a good convergent validity for each dimension of the MMHLS [46]. In addition, the discriminant validity was identified by the comparison of the square root of the AVE and the correlations between the three dimensions, where a value of the square root of AVE higher than the correlation coefficient indicated a good discriminant validity between different dimensions [47]. Furthermore, most researchers apply Rasch analysis for examining instrument quality and it has been suggested that using factor analysis and Rasch analysis can evaluate instrument quality more comprehensively [48,49,50,51]. Thus, we conducted multidimensional Rasch modeling, performed in ConQuest 2.0 (ACER, Victoria, Australia), due to the MMHLS being a multidimensional construct. Indicators of item fit statistics (i.e., Infit MNSQ and Outfit MNSQ) were used to examine the model–data fit. In addition, since gender was found to have a substantial impact on mental health literacy, the evidence for a differential item function (DIF) between genders was identified. Furthermore, reliability was also checked using Rasch reliabilities.

## 3. Results

### 3.1. Preliminary Analysis

Before the formal investigation, fifteen elite athletes were recruited to give feedback on the MMHLS items through face-to-face conversations. This sample consisted of seven female and eight male elite athletes with a mean age of 20.9 years (SD = 2.1, range 18–24). The readability and understandability of the items were confirmed by all participants, while only a few grammatic typos were corrected based on participants’ feedback. The results of the pilot test indicated that some athletes were confused by item K6 (Cognitive behavioral therapy can change the way a person thinks and reacts to stress) and K7 (A person with bipolar disorder may show a dramatic change in mood) because those two items included professional psychological terms (i.e., cognitive behavioral therapy, bipolar disorder). All 26 items remained in the final version—including these two items, which accessed the participant’s knowledge of symptoms and mental illness; there was an emphasis in the instructions to please select the “I don’t know” option if a participant was not familiar with this content.

### 3.2. Factorial Validity

As MHL is conceptualized as a higher-order factor affecting knowledge-oriented MHL, belief-oriented MHL, and resource-oriented MHL [15], we examined the model fit with a second-order factorial model—the results are displayed in Table 1. Although the value of the model fit on the initial model was satisfactory and at an acceptable level, two items showed poor function. Specifically, the factor loading of item K9 “When a person stops taking care of his or her appearance, it may be a sign of depression” was 0.22 points lower than the acceptable level of 0.40, and the factor loading of item B6 “Poor parenting cause schizophrenia” was not significant (*p* = 0.08 > 0.05). The model fit index appeared better after removing these two items (χ^2^/*df* = 1.465, CFI = 0.955, TLI = 0.950, RMSEA = 0.038). In addition, all the factor loadings were significant, ranging from 0.43 to 0.93 (>0.40; see Figure 1).

### 3.3. Convergent Validity and Discriminant Validity

The value of the Average Variance Extracted (AVE) and Composite Reliability (CR) were assessed to determine the convergent validity [52,53]. As shown in Table 2, all scores of CR were satisfied with the rule of thumb (0.60) and were higher than the AVE. However, only the MHL-Resources’ AVE was higher than 0.50, and the values for the others were lower than 0.50. Besides this, all the square root AVE values were higher than the correlations between the two dimensions, which indicated that the discriminant validity of the data was good.

### 3.4. Concurrent Validity

The results indicated that the MMHLS had a significantly positive correlation with the ATSPPH-SF—attitudes toward professional Psychological Help Scale (*r* = 0.23) and a negative correlation with the SSPPH—the stigma against seeking professional psychological help (*r* = −0.23; see Table 3 for details). Further analyses examined the differences between genders and educational levels in these three modified factors of the MMHL in the elite athlete sample. There was a significant difference between genders with respect to both the Knowledge-oriented MHL (t (318) = 3.35, *p* < 0.05) and the Beliefs-oriented MHL (*t*_318_ = 4.62, *p* < 0.01), whereby female athletes had significantly higher literacy than males. However, for educational level, no significant differences were discovered. Factor scores are shown in Table 4.

### 3.5. Internal-Consistency Reliability and Test–Retest Reliability

Descriptive statistics as to the means, standard deviations, and internal-consistency reliability of the MMHLS for the samples are summarized in Table 3. In terms of the internal-consistency reliability, the MMHLS exhibited an adequate Cronbach’s alpha (range = 0.75–0.78; 0.81 for the total scale) and composite reliability coefficients (range = 0.87–0.92). Test–retest reliability was examined after one month for elite athletes (*n* = 95), reaching 0.66.

### 3.6. Rasch Analysis

A Rasch analysis was applied to the 24 items confirmed by a factor analysis on the dataset. We employed a Rasch’s simple logistic model for dichotomies [54]. The results showed that all the items had a satisfactory fit to the Rasch model, i.e., item fit scores ranged from 0.80 to 1.24 (Infit MNSQ) and 0.69 to 1.48 (Infit MNSQ), falling within the acceptable range of 0.6 to 1.5 [55]. This finding indicates that the latent trait was theoretically measured by all items. The DIF was identified by evaluating the differences in item difficulty across groups after controlling the latent trait levels. Conrad et al. has suggested that a differences of item difficulty larger than 0.6 logits between groups are considered to have a substantial DIF [56]. The analysis found that three items displayed poor DIFs across gender, and Rasch reliabilities for the three subscales were 0.73, 0.75 and 0.75, respectively.

## 4. Discussion

This study first examined the factorial validity, convergent and discriminant validity, concurrent validity, internal-consistency reliability, and test–retest reliability of the MMHLS for use with Chinese elite athletes. The final MMHLS contained three dimensions: knowledge, beliefs, and resources across 24 items for the elite athlete. The results showed good psychometric properties for the MMHLS in Chinese elite athletes.

The factorial structure of the MMHLS [15,29] was identified and replicated in this study. Two poor items were removed in the final scale. Item K9 (When a person stops taking care of their appearance, it may be a sign of depression) had cross-loading with another two factors (i.e., MHL-Resources and second-order MHL), and the standard factor loading with Knowledge was lower than the cutoff 0.40 (*λ* = 0.23). Item B6 (Poor parenting cause schizophrenia), which has been found problematic in previous studies [31], also showed poor function. Nevertheless, the three-factor model supports the theoretical assumption that MHL is a multi-component concept [14]. It also involved items regarding mental health resources identified as crucial information for help-seeking or helping someone with mental disorders in real life [29].

For convergent validity, there were some minor concerns observed in the results. The AVE for both the MHL-Knowledge oriented subscale and the MHL-Beliefs oriented subscale did not meet the criterion. However, Malhotra and Dash suggested that it is strict and conservative to apply AVE to measure convergent validity and that CR might be a good indicator for convergent validity assessment [57]. Therefore, the MMHLS shows adequate convergent validity and discriminant validity for the current sample. In terms of correlations with other related scales, although there was a good pattern of relationships between mental health literacy and attitudes toward seeking professional psychological help, the correlation coefficient was weak compared to previous studies [19,58]. The potential reasons for this might be a lack of time, little support from the sports environment, and stigma.

Regarding the reliability, the value of the Cronbach’s alpha for the total scale and the composite reliability for the subscales had an adequate internal-consistency reliability for the second-order three-dimensional MMHLS in elite athletes. Given the limitations of such approaches [29,30], the test–retest reliability was also examined for the MMHLS, which confirmed the previous study’s findings [31]. However, the results for the test–retest reliability just reaching an acceptable level might be due to the long intervals caused by the competition schedule and the COVID-19 pandemic.

For concurrent validity, the relationship between the MMHLS and attitudes toward help-seeking and stigma was in line with previous studies [17,19]. In addition, analyses indicated that female elite athletes reported higher MHL levels than male athletes. This confirms the findings of previous studies, which have found that females are more likely to have higher levels of mental disorders such as anxiety disorders and depression than males because of their higher prevalence in women and, therefore, they have a greater awareness of the symptoms [21,30]. There was no significant difference in mental health literacy between individuals in different years of the study, which is consistent with previous research [59].

Overall, the psychometric strength of the MMHLS has been demonstrated in this study. Nevertheless, several limitations need to be noted: First, the data in this study were from a convenience sample from one sports training center. The use of convenience sampling introduces self-selection bias. Therefore, a stratified sampling approach is desirable in the future. Second, although three items performed poorly in DIF function across gender, considering the importance of those items and the initial examination in elite athletes, we still reserved these items; measurement invariance across genders, age groups, and sports levels needs to be identified in the future. Furthermore, all variables were taken through self-description questionnaires for evaluation, so the results may be biased. As a result, different methods and objective criteria are necessary.

Our study only provided initial evidence for the factorial validity, convergent and discriminant validity, concurrent validity, internal-consistency reliability, and test–retest reliability of the MMHLS. Future studies could go further in establishing more robust evidence for predictive validity and structural validity. As mental health literacy was conceptualized to be particularly affected in sports environments [60], it is also significant that the assessment is validated across a wider variety of contexts—also including more organization and management factors.

## 5. Conclusions

In conclusion, this study provides support for the factorial validity, convergent and discriminant validity, concurrent validity, internal-consistency reliability, and test–retest reliability of the 24-item three-dimensional MMHLS for use with Chinese elite athletes. The 24-item three-dimensional MMHLS can be regarded as a psychometrically sound instrument for evaluating mental health literacy in Chinese elite athletes, which can be used to contribute to future observational and experimental research on mental health literacy and mental health in elite athletes.

## Figures and Tables

**Figure 1 ijerph-19-12620-f001:**
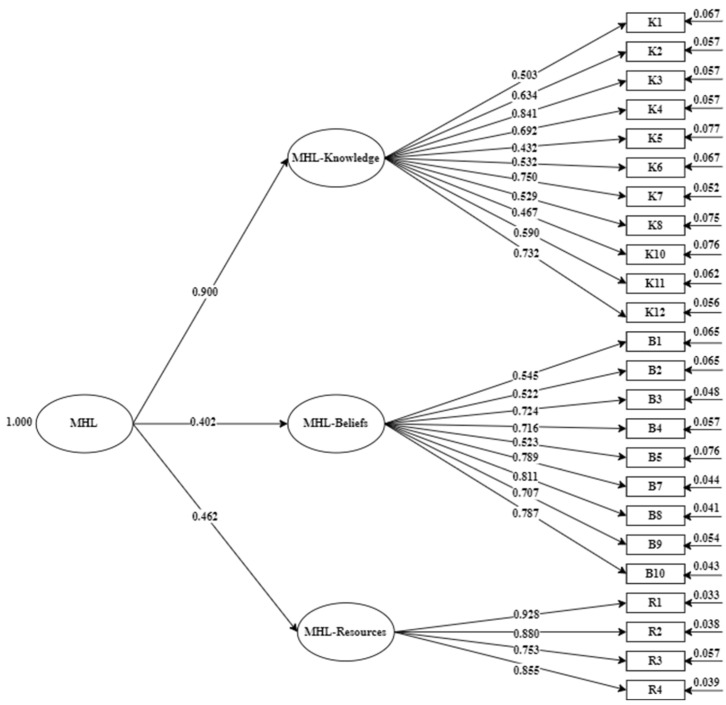
Second-order model with standardized factor loading on MHL-Knowledge, MHL-Beliefs, and MHL-Resources (*n* = 320).

**Table 1 ijerph-19-12620-t001:** Model fit indices of the second-order three-dimensional MMHLS in elite athletes (*n* = 320).

Model	*χ* ^2^	*df*	χ^2^/*df*	CFI	TLI	RMSEA	RMSEA 90%CI
Model	473.062	296	1.598	0.930	0.923	0.043	0.036–0.050
Model a	364.813	249	1.465	0.955	0.950	0.038	0.029–0.046

Note: χ2 = Chi square; df = degree of freedom; CFI = goodness-of-fit index; TLI = Tucker–Lewis index; RMSEA = root mean square error of approximation; CI = confidence interval; a = model with modification.

**Table 2 ijerph-19-12620-t002:** Convergent validity, discriminant validity, and reliability.

Dimension	Item	M	SD	α	CR	AVE	1	2	3
1. MHL-Knowledge	11	5.35	2.74	0.75	0.87	0.39	0.62 *		
2. MHL-Beliefs	9	3.98	2.58	0.78	0.89	0.46	0.36	0.69 *	
3. MHL-Resources	4	1.79	1.49	0.78	0.92	0.73	0.42	0.19	0.86 *

Note: M = Mean, SD = Standard deviation, CR = Composite Reliability, AVE = average variance extracted, * = the square root of the AVE.

**Table 3 ijerph-19-12620-t003:** Correlation between MMHLS and related variables.

	ATSPPH-SF	SSPPH
MHL-Knowledge	0.17 **	−0.18 **
MHL-Belief	0.17 **	−0.14 *
MHL-Resource	0.16 **	−0.20 **
Total MMHLS	0.23 **	−0.23 **

Note: MHL = Mental Health Literacy; MMHLS = Multicomponent Mental Health Literacy Scale; ATSPPH-SF = Attitudes toward Seeking Professional Psychological Help Scale–Short Form; SSPPH = Questionnaire of Stigma for Seeking Professional Psychological Help; ** < 0.01, * < 0.05.

**Table 4 ijerph-19-12620-t004:** Mental Health Literacy scores by subconstruct.

	*n*	Knowledge	Beliefs	Resources
		M(SD)	M(SD)	M(SD)
Male	183	4.90 (2.69)	3.42 (2.59)	1.73 (1.53)
Female	138	5.94 (2.72)	4.72 (2.39)	1.86 (1.42)
High school	111	5.28 (2.63)	3.83 (2.45)	1.84 (1.51)
University	209	5.38 (2.81)	4.06 (2.65)	1.76 (1.47)

Note: Scores on Knowledge-oriented, Beliefs-oriented, and Resources-oriented beliefs can range from 0–11, 0–10, and 0–4, respectively. Higher scores on all scales denote greater mental health literacy.

## Data Availability

The datasets generated during and/or analyzed during the current study are available from the corresponding author on reasonable request.

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
