# Peer review of "Further Examination of the Psychometric Properties of the Multicomponent Mental Health Literacy Scale: Evidence from Chinese Elite Athletes"

_ijerph, 2022, doi:10.3390/ijerph191912620_

Round 1
Reviewer 1 Report
Dear Authors;
Article Review:
Strong points:
The article presents a relevant topic given the mental health problems reported in the literature in elite athletes. In the introduction, the importance of this investigation is justified. The methodology and statistical analysis are adequate to the study objectives.
Points to improve:
- Authors should better explain how participants were recruited for the study;
- in the description of the procedures it is not clear whether the participants signed an informed consent to participate in the research
- the pilot study needs some more details, how the feedback was received from the participants (interview? ; conversation?).
- On line 163, authors must write (n= 222; 69%);
- the authors present results with decimals and results without decimals (examples: line 203; line 209; line 275; line 276; Table 3). Thus, authors should standardize the way in which the results are presented.
- the conclusions should be further developed, explaining the practical implications of adapting the instrument to the intervention and to the investigation carried out with elite athletes.
For the article to be published, the points mentioned must be improved.
Author Response
Response: Thanks a lot for your appreciation of our study. We have carefully read each comment and responded as below. We have also revised the manuscript accordingly.
1.Authors should better explain how participants were recruited for the study.
Response: Thank you for this suggestion. We have elaborated more on the participant recruitment procedure (p. 4; Line 166-172).
“The Guangdong provincial sports training center is a cooperation agency in this project. And several sports psychologists in this article have provided long-term service for different teams in this center. To ensure a smooth investigation, the purpose and nature of the study was explained to team leaders and coaches before requesting permission to recruit participants. Upon their approval, the recruitment information was verbally disseminated to the athletes by the sport psychology interns with the assistance of team coaches.”
2. in the description of the procedures it is not clear whether the participants signed an informed consent to participate in the research.
Response: Thanks for pointing out this issue. Yes, all the participants signed the written informed consent form before completing the research. We have added this information accordingly (p. 4; Line 174-177).
“All the participants signed the written informed consent form before the study commencement. They completed the survey independently and voluntarily and were allowed to withdraw from the study at any time without any negative consequences.”
3.the pilot study needs some more details, how the feedback was received from the participants (interview? ; conversation?).
Response: Thanks for pointing this out. We have elaborated more on this point (p. 5; Line 250-252).
“Before the formal investigation, fifteen elite athletes were recruited to give feedback on MMHL items through face-to-face conversation. This simple consisted of 7 female and 8 male elite athletes with a mean age of 20.9 years (SD = 2.1, range 18-24)”
4.On line 163, authors must write (n= 222; 69%).
Response: Thank you for pointing this out. We have corrected this point accordingly.
5.the authors present results with decimals and results without decimals (examples: line 203; line 209; line 275; line 276; Table 3). Thus, authors should standardize the way in which the results are presented.
Response: Thank you for pointing this out. We have corrected it accordingly.
Table 1. Model fit indices of the second-order with a three-factor MMHL model among elite athletes.
|
Model |
χ2 |
df |
χ2/df |
CFI |
TLI |
RMSEA |
RMSEA 90%CI |
|
Model |
473.062 |
296 |
1.598 |
.930 |
.923 |
.043 |
.036-.050 |
|
Model a |
364.813 |
249 |
1.465 |
.955 |
.950 |
.038 |
.029-.046 |
|
Note: χ2 = Chi square; df =degree freedom; CFI = goodness-of-fit index; TLI =Tucker-Lewis index; RMSEA = root mean square error of approximation; CI = confidence interval; a = model with modification. |
|||||||
Table 2. Convergent validity, discriminant validity, reliability.
|
Dimension |
Item |
M |
SD |
α |
CR |
AVE |
1 |
2 |
3 |
|
1. MHL-Knowledge |
11 |
5.35 |
2.74 |
.75 |
.87 |
.39 |
.62* |
|
|
|
2. MHL-Beliefs |
9 |
3.98 |
2.58 |
.78 |
.89 |
.46 |
.36 |
.69* |
|
|
3. MHL-Resources |
4 |
1.79 |
1.49 |
.78 |
.92 |
.73 |
.42 |
.19 |
.86* |
|
Note: M = Mean, SD = Standard deviation, CR = Composite Reliability, AVE = average variance extracted, * = the square root of AVE. |
|||||||||
Table 3: correlation between MMHL with related variables
|
|
ATSPPH-SF |
SSPPH |
|
MHL-Knowledge |
.17** |
-.18** |
|
MHL-Belief |
.17** |
-.14* |
|
MHL-Resource |
.16** |
-.20** |
|
Total MMHL |
.23** |
-.23** |
|
Note: MHL = Multicomponent Mental Health Literacy; ATSPPH-SF = Attitudes toward Seeking Professional Psychological Help-Seeking-Short Form; SSPPH = Questionnaire of Stigma for Seeking Professional Psychological Help; ** <.01,* <.05; “-” = not significant. |
||
6.the conclusions should be further developed, explaining the practical implications of adapting the instrument to the intervention and to the investigation carried out with elite athletes.
Response: Thank you for pointing this out. We have improved the conclusions accordingly (p. 10; Line 382-388).
Now it reads as:
“In conclusion, this study provides support for the factorial validity, convergent and discriminant validity, concurrent validity, internal-consistency reliability, and test-retest reliability of the 24-item three-dimensional MMHL for use with Chinese elite athletes. The 24-item three-dimensional MMHL can be regarded as a psychometrically sound instrument for evaluating mental health literacy in Chinese elite athletes, which can be used and contribute to the future observational and experimental research on mental health literacy and mental health among elite athletes.”
Reviewer 2 Report
The article provides a thorough introduction with a very good presentation of the literature and the rationale for the study.
The following are concerns with the study:
a) The authors mention a pilot study but this was not clearly described in the paper.
b) In the section of construct validity, in fact, it is factorial validity that it is tested. This should be corrected throughout the manuscript.
c) The evaluation of convergent and discriminant validity does not seem to be appropriate. Convergent validity takes two measures that are supposed to be measuring the same construct and shows that they are related. Conversely, discriminant validity shows that two measures that are not supposed to be related are in fact, unrelated. The average variance extracted has often been used to assess discriminant validity but perhaps is not the most appropriate way.
d) In fact convergent and discriminant validity are indicators of construct validity.
e) In the convergent/discriminant validity section the scores reported are not consistent; e.g., 'The AVE needs to be > .50 ' and 'AVE was higher than .05'. Consider clarifying.
f) The need to perform Rasch analysis has not been fully justified. What this analysis offers to the paper.
Author Response
Response: Thanks a lot for your appreciation of our study. We have carefully read each comment and responded as below. We have also revised the manuscript accordingly.
1.The authors mention a pilot study but this was not clearly described in the paper.
Response: Thanks for pointing this out. We have elaborated more on this point (p. 5; Line 250-252)
“Before the formal investigation, fifteen elite athletes were recruited to give feedback on MMHL items through face-to-face conversation. This simple consisted of 7 female and 8 male elite athletes with a mean age of 20.9 years (SD = 2.1, range 18-24)”
2. In the section of construct validity, in fact, it is factorial validity that it is tested. This should be corrected throughout the manuscript.
3. In fact, convergent and discriminant validity are indicators of construct validity.
Response: Thank you for pointing this out. We have corrected this point and used factorial validity throughout the manuscript.
4. The evaluation of convergent and discriminant validity does not seem to be appropriate. Convergent validity takes two measures that are supposed to be measuring the same construct and shows that they are related. Conversely, discriminant validity shows that two measures that are not supposed to be related are in fact, unrelated. The average variance extracted has often been used to assess discriminant validity but perhaps is not the most appropriate way.
Response: Thank you for your comments. We totally agree that at a whole scale level, convergent validity takes two measures that are supposed to be measuring the same construct and show that they are related. Conversely, discriminant validity shows that two measures that are not supposed to be related are in fact, unrelated (Taherdoost, 2016). However, the convergent and discriminant validities can be also examined at the dimensional/sub-scale level, which is also a common practice for the scales with multidimensions/sub-scales (Linn, 2011; Almén et al., 2018; Anlı, 2019; Lee, 2019). In this study, we used convergent validity to test each item within a dimension measuring the same construct. While the discriminant validity was used to examine each dimension discriminates from other dimensions (i.e., knowledge-oriented MHL, belief-oriented MHL and resource-oriented MHL).
To avoid confusing the readers, we have elaborated more on this point in the statistical analysis part. (p.5; Line 230-239)
“As the MMHL is a three-dimensional scale, the convergent validity was examined for each dimension while the discriminant validity was examined between the three dimensions of MMHL (i.e., knowledge-oriented MHL, belief-oriented MHL and resource-oriented MHL). The factor loadings were employed to calculate the average variance extracted (AVE) and composite reliability (CR), where AVE > .50 and CR > .60 indicates a good convergent validity for each dimension of MMHL [46]. In addition, the discriminant validity was identified by the comparison of the square root of AVE and correlations between the three dimensions, where s higher value of the square root of AVE than the correlation coefficient indicates a good discriminant validity between different dimensions [47].”
Reference:
Linn, R. L. (2011). The standards for educational and psychological testing: Guidance in test development. In Handbook of test development. Routledge.
Taherdoost, H. (2016). Validity and reliability of the research instrument; how to test the validation of a questionnaire/survey in a research. How to test the validation of a questionnaire/survey in a research (August 10, 2016).
Lee, D. (2019). The convergent, discriminant, and nomological validity of the Depression Anxiety Stress Scales-21 (DASS-21). Journal of affective disorders, 259, 136-142.
Almén, N., Lundberg, H., Sundin, Ö., & Jansson, B. (2018). The reliability and factorial validity of the Swedish version of the Recovery Experience Questionnaire. Nordic Psychology, 70(4), 324-333.
Anlı, G. (2019). Adaptation of the prosocial behavioral intentions scale for use with Turkish participants: Assessments of validity and reliability. Current Psychology, 38(4), 950-958.
- Liu, H-X.; Chow, B-C.; Liang, W.; Hassel, H.; Huang, Y. W. Measuring a broad spectrum of eHealth skills in the Web 3.0 context using an eHealth Literacy Scale: Development and validation study. Journal of Medical Internet Research. 2021, 23, e31627.
- Franke, G.; Sarstedt, M. Heuristics versus statistics in discriminant validity testing: a comparison of four procedures. Internet Research. 2019.
5. In the convergent/discriminant validity section the scores reported are not consistent; e.g., 'The AVE needs to be > .50 ' and 'AVE was higher than .05'. Consider clarifying.
Response: Thank you for pointing this typo. We have revised it accordingly (p.7; Line 285-286).
“However, only MHL-Resources' AVE was higher than .50, and the figures of others were lower than .50.
6. The need to perform Rasch analysis has not been fully justified. What this analysis offers to the paper.
Response: Thank you for this suggestion. We have rephrased this part accordingly (p. 6; Line 239-247).
“Furthermore, most researchers applied Rasch analysis for examining instrument quality and they have suggested that using factor analysis and Rasch analysis can evaluate instrument quality more comprehensively [46-49]. Thus, we conducted multidimensional Rasch model performed in ConQuest 2.0 due to MMHL is a multidimensional construct. Indicator of item fit statistics (i.e., Infit MNSQ and Outfit MNSQ) was used to test the model-data fit. In addition, since gender was found to have a substantial impact on mental health literacy, the evidence for differential item function (DIF) across gender was identified. Furthermore, reliability will also be checked through Rasch reliabilities.”.
Reference:
- 46. Wang, X.; Yan, Z.; Huang, Y.; Tang, A.; Chen, J. Re-Developing the Adversity Response Profile for Chinese University Students. International Journal of Environmental Research Public Health. 2022, 19, 6389.
- Deneen, C.; Brown, G.T.; Bond, T.G.; Shroff, R. Understanding outcome-based education changes in teacher education: Evaluation of a new instrument with preliminary findings. Asia-Pac. J. Teach. Educ. 2013, 41, 441–456.
- Testa, I.; Capasso, G.; Colantonio, A.; Galano, S.; Marzoli, I.; di Uccio, U.S.; Trani, F.; Zappia, A. Development and validation of a university students’ progression in learning quantum mechanics through exploratory factor analysis and Rasch analysis. International Journal of Science Education. 2019, 41, 388–417.
- Yan, Z.; Brubacher, S.; Boud, D.; Powell, M. Psychometric properties of the self-assessment practice scale for professional training contexts: Evidence from confirmatory factor analysis and Rasch analysis. International Journal of Training and Development. 2020, 24, 357–373.
Round 2
Reviewer 2 Report
The authors have addressed the comments and the manuscript has been improved.